# Acute Effects of a Percussive Massage Treatment on Movement Velocity during Resistance Training

**DOI:** 10.3390/ijerph18157726

**Published:** 2021-07-21

**Authors:** Manuel García-Sillero, Jose Manuel Jurado-Castro, Javier Benítez-Porres, Salvador Vargas-Molina

**Affiliations:** 1Faculty of Sport Sciences, EADE-University of Wales Trinity Saint David, 29018 Málaga, Spain; salvadorvargas@eade.es; 2Metabolism and Investigation Unit, Maimonides Biomedical Research Institute of Cordoba (IMIBIC), Reina Sofia University Hospital, University of Cordoba, 14004 Córdoba, Spain; juradox@gmail.com; 3Faculty of Medicine, University of Málaga, 29071 Málaga, Spain; benitez@uma.es

**Keywords:** performance, musculoskeletal manipulations, fatigue, velocity loss, recovery, therapy

## Abstract

The aim of this research was to verify whether the application of percussion therapy during inter-set rest periods increases the number of repetitions performed before reaching a 30% velocity loss threshold during a bench press exercise. **Methods:** Twenty-four male university students participated in this study (24.3 ± 1.3 years; 77.5 ± 8.3 kg; 177.0 ± 5.6 cm; 24.7 ± 2.6 kg∙m^−2^). Participants were randomized into two groups: a percussion therapy group (PTG) and a control group (CG). They performed 4 sets at 70% of a one-repetition maximum before reaching a 30% velocity loss threshold with an inter-set recovery of 3 min. **Results**: The PTG performed a greater total number of repetitions compared to the CG (44.6 ± 4.8 vs. 39.5 ± 6.8; *p* = 0.047; ES = 0.867). No differences were observed for the different movement velocity variables and fatigue control (*p* > 0.05). **Conclusions**: Percussion therapy is an effective method to delay the loss of movement velocity in the bench press exercise.

## 1. Introduction

Velocity-based training (VBT) is a contemporary method of resistance training (RT) that enables accurate and objective evaluation or prescription of RT intensities and volumes [1].

It is well known that movement velocity is a very stable variable for an athlete, although performance varies, especially between people with very different performance levels [2]. The movement velocity has been described as an optimal indicator of an athlete’s state of physical fatigue [3]. Changes in mean velocity (MV) may be indicative of altered neuromuscular qualities [4]. Reductions in velocity may be symptomatic of fatigue, overload/overtraining, or detraining/misalignment, while faster velocities could mean improvements in neuromuscular capacity or acute potentiation [4,5]. The literature contains indicators of fatigue in RT, such as the loss of velocity as an indicator of accumulated fatigue during the sets, and the effort index (EI), which indicates the relationship between what is done and what could potentially be done [6].

In order to stimulate specific adaptations, it has been proposed to stop training exercises when a certain threshold of velocity loss is reached [7]. In this sense, completing a greater number of repetitions before reaching that threshold may trigger greater adaptations [8]. Therefore, coaches may look for intra-set strategies, such as cluster configuration [9] and inter-set strategies [10], that allow athletes to perform more repetitions before reaching a specified threshold of velocity loss.

In addition, different strategies have been proposed to optimize post-exercise recovery and [11] and inter-set recovery during RT; some are based on seeking an optimal recovery time [12], while other aspects related to physical therapy, such as massage, have been proposed for recovery in RT [13]. Recently, the application of foam roller self-massage during lower extremity exercise inter-set rests has been investigated, with findings indicating that the use of this technique hinders performance and increases perceived exertion [14]. In contrast, handheld percussive massage treatment (PT) has gained popularity in the therapeutic and athletic communities in recent years [15,16]. Nowadays, many different manufacturers provide percussion devices for both self-massage and massage by a therapist. The modality of PT is possible due to advancements in mechanical engineering, which allow devices to elicit compressive forces at a set frequency, with this slowing down as more force is applied by the user. PT is used in the treatment of deep tissue, with benefits including pain reduction, increased blood flow, improved scar tissue, decreased lactate, reduced muscle spasms, increased lymphatic flow, inhibition of the Golgi reflex, increased range of motion, and better recovery based on the principles of fascial connective tissue treatment [17]. Although the research on PT application for sports performance is very limited, the effects of PT on jumping ability have recently been evaluated in young athletes as a warm-up, showing no beneficial effect on jumping height [18]. Another study showed that PT improved the range of motion in dorsiflexion without affecting muscle strength [15], after a 5 min massage treatment. However, PT interventions have shown a potential effect in restoring muscle compliance and reducing stiffness, creating an optimum environment for muscle recovery between sets during RT, due to a possible optimization of recovery of muscle tissue [16]. In relation to these aspects, PT could be a valid strategy to delay RT fatigue. Thus, due to the benefits of PT in muscle tissue and the lack of research on its acute effects on inter-set recovery during RT, this study aimed to evaluate the effect of PT application on movement velocity in a bench press (BP) exercise during RT, objectively evaluating the possible fatigue reduction generated and the EI. We hypothesized that the use of PT during rests between RT sets may delay the onset of fatigue and improve performance.

## 2. Materials and Methods

### 2.1. Trial Design

This was a randomized controlled pilot study on resistance-trained men. The study was designed following the Consolidated Standards of Reporting Trials (CONSORT) (CONSORT checklist—Appendix A). The first session was used for body composition assessment and familiarization with test protocols. The participants arrived at the laboratory in the morning, rested and fasting. After determining their body composition, height (SECA 220, Hamburg, Germany), and total mass (Tanita RD-545, Tokyo, Japan), they performed the BP exercise with light and moderate loads, and the researchers emphasized the correct technique. The day after, a progressive load test was used to determine the various load–speed relationships and one-repetition maximum (1RM) strength. This involved 5 min of static stretching and upper body mobilization exercises, and then a set of 5 repetitions of BP, with a fixed load of 20 kg. Each participant was instructed to lower the barbell to the chest above the nipples in a slow and controlled manner, and then wait there in an alert state until instructed to lift by an experienced evaluator. A pause lasting about 1.5 s was applied between the eccentric phase and the concentric phase to minimize the impact of the rebound effect and achieve more repeatable and consistent measurements [19]. The participants were not allowed to bounce the barbell from their chest or lift their shoulders or torso off the bench. In each repetition, strong verbal encouragement and velocity feedback was provided to motivate participants to give their maximum effort. Each participant was instructed to always perform each repetition in an explosive manner, and push the barbell up from the chest as soon as possible after hearing the ‘start’ command. Based on current evidence, MV was used as an indicator throughout the sets [20].

The initial load of all participants was set to 20 kg and gradually increased in 10 kg increments until the MV reached was less than 0.5 m·s^−1^. Thereafter, the load was adjusted in smaller increments (5–10 kg) for each participant, so that the 1RM could be determined accurately. The heaviest load at which each participant could correctly lift with full elbow extension was considered his 1RM. To ensure safety when lifting very heavy weights, there were well-trained trainers on both sides of the barbell. For lighter loads (MV > 1.0 m·s^−1^), 3 attempts were performed for each load. There were two attempts with medium loads (0.65 m·s^−1^ ≤ MV ≤ 1.0 m·s^−1^); and only one for the maximum load (MV < 0.65 m·s^−1^).

The rest time was 2–3 min for light and medium loads, and 5–6 min for heavy loads. According to the standard of the fastest MV, only the best repetition of each load was considered for analysis [21].

### 2.2. Participants

A group of convenience-sampled students selected from the available population at EADE University (Wales Trinity Saint David University, Malaga, Spain) was potentially eligible. A total of 24 university students participated in this study (24.3 ± 1.3 years; 77.5 ± 8.3 kg; 177.0 ± 5.6 cm; 24.7 ± 2.6 kg∙m^−2^).

Students of Physical Activity Sciences with no previous upper limb injuries during the 6 months prior to the study and with more than 2 years of RT experience were included. The participants abstained from intense upper limb exercise for 24 h prior to the start of the familiarization sessions until the completion of the assessments. Additionally, they were asked not to take any dietary supplements or medications during the experimental period, especially any type of stimulant substance. The participants were informed of the possible harmful risks of the experiment and provided written informed consent agreeing to the conditions of the study. The research protocol was reviewed and approved by the Ethics Committee of the EADE, University of Wales Trinity Saint David (Málaga, Spain) Committee (reference number: EADECAFYD2020-4), and was conducted in accordance with the guidelines of the Declaration of Helsinki.

### 2.3. Procedures

#### 2.3.1. Bench Press Exercise

The participants performed the third session 72 hours after the RM test. A total of 4 BP sets were performed with a load of 70% of a 1RM [3], and participants were instructed to execute each repetition at the maximum possible velocity. For fatigue management, the decline in repetition velocity during the three consecutive exercise sets was monitored. It was calculated as the percentage loss in MV from the fastest (usually first) to the slowest (last) repetition of each set, and averaged over the four sets using the linear encoder SmartCoachTM (SmartCoach Europe AB, Stockholm, Sweden; SC).

The total number of repetitions performed by the participants was calculated, allowing a velocity loss threshold of 30% in each set, with an inter-set recovery of 3 min [22]. In addition, the EI was an indicator of accumulated fatigue and the level of effort for each condition (PTG and CG) was calculated [23].

#### 2.3.2. Percussion Therapy

The Theragun^®^ G3 Pro (Therabody, Los Angeles, CA, USA) device was used for the experimental treatment in the PTG. The PT treatment provided by the device during this study had the following mechanical characteristics: amplitude (16 mm), torque (60 pounds), and frequency (2400 per minute). PT was applied to each participant immediately following completion of the last rep at the end of each set. PT was applied to the pectoralis major and minor, given that the standardized grip used in our study was 100% or more of the biacromial width [24], and the bench had no inclination (0°) [25], with the pectoral as the muscle group with the highest activation during the BP exercise. PT was applied to the muscle in the PTG with the dampener attachment using moderate force and fast movement, gliding up and down along the muscle belly from the origin to the insertion for 15 s, ensuring constant pressure at all times, and following the direction of the muscle fibers. This was applied on both pectoralis muscles, around the medial half of the anterior border of the clavicle, anterior aspect of the sternum, first 6 costal cartilages, and aponeurosis of the external oblique. The CG performed its four series without the application of PT.

### 2.4. Randomization of Participants

The students were randomized by permute block (http://www.randomization.com) (accessed on 19 March 2021) into two groups: the PT group (PTG), *n* = 12, and the control group (CG), *n* = 12.

### 2.5. Statistical Methods

To study the normality of the variables, Shapiro–Wilk tests were performed. A comparison of the mean outcomes (number of repetitions and movement velocity variables) between the two groups (PTG vs. CG) was conducted using a Student’s *t*-test. The effect size (ES) was calculated following the method developed by Lakens [26], using Cohen’s d as the ES index, where ES can be roughly classified as small (<0.2), medium (0.5 to 0.8), and large (>0.8) [27]. In addition, a general linear model for repeated measures (GLM-RM) was applied for the effect of time (different sets analyzed), condition (PTG vs. CG), and the time–condition interaction on the total number of repetitions performed. The Greenhouse–Geisser adjustment for sphericity was calculated. After a significant F-test, differences among the means were identified using pairwise comparisons with Bonferroni’s adjustment. GLM-RM ES were calculated using partial eta squared ($\eta_{p}^{2}$), considering small to be under 0.25, medium 0.26–0.63, and large above 0.63 [28]. The data are displayed as the mean difference (MD) ± standard deviation (SD), or comparing the mean ± SD for each condition, continued from ES. Significance was set at *p* < 0.05 or *p* < 0.01. SPSS software v25 (IBM, Portsmouth, United Kingdom) was used for the statistical analysis.

## 3. Results

Figure 1 presents a diagram of subject enrollment as per the guidelines of the Consolidated Standard of Reporting Trials (CONSORT).

Table 1 shows the characteristics of the participants in each group. There were no differences in the load (kg) lifted for each group (PTG: 52.5 ± 11.2 kg vs. CG: 55.2 ± 13.4 kg).

Significant differences were observed in Set 3 (*p* = 0.007), but no differences were observed for other sets (Set 1: *p* = 0.878; Set 2: *p* = 0.156; Set 4: *p* = 0.134) (Table 2). The PTG performed a greater total number of repetitions compared to the CG (*p* = 0.047).

According to the GLM-RM, differences in time were found in the number of repetitions (F = 6.997, *p* = 0.002, $\eta_{p}^{2}$ = 0.950). However, there was a non-statistically significant effect for the time–condition interaction on the number of repetitions (F = 1.788, *p* = 0.182, $\eta_{p}^{2}$ = 0.394). It was observed that the PTG maintained the ability to perform a similar number of repetitions from Set 1 to Set 4, without differences (MD: 1.1 ± 0.57; *p* = 0.084). However, the CG obtained differences, with a reduction in the repetitions performed (MD: 3 ± 0.9; *p* = 0.007).

No differences were observed between the PTG and CG for the different variables obtained (repetition with maximum velocity; peak velocity; peak power) by controlling the movement velocity for the BP exercise (Table 3).

No differences were observed for the EI (PGT: 18.3 ± 3.2 vs. CG: 18.5 ± 2.7; *p* = 0.848; ES = 0.079).

## 4. Discussion

The aim of this study was to evaluate the reduction of fatigue through the application of PT in the BP exercise. Although the results of this study did not show differences in the movement velocity variables (repetition with maximum velocity; peak velocity; peak power), there was an improved response in muscular endurance, with a greater number of repetitions in the PTG compared to the CG, with no difference between groups in the EI (*p* = 0.848). EI is used as an independent variable when analyzing training effects [6]. This is obtained from the maximum velocity at which the load is applied and the velocity loss in performing repetitions or sets. This finding may be of great application in the field of RT. Using the velocity loss in the set as a tool for prescribing and monitoring the amount of RT instead of specifying a fixed number of repetitions for a given load seems to be a more reasonable representation of resistance exercise stimulation [7]. These results suggest that PT is an effective strategy for delaying fatigue, thereby improving muscular performance in upper limb strength training.

The key finding of the present study is the significant increase in the number of repetitions performed over the consecutive sets for the BP exercise when applying PT in the inter-set period, compared with a passive recovery interval. Although there were only significant differences in Set 3, the PT group performed a greater number of repetitions in Sets 2, 3, and 4. A total of 5 repetitions more with maximum velocity intention is a clear possibility of improved performance, taking into account the specific advantages of training with maximum velocity intention in the BP exercise [29]. Moreover, according to the GLM-RM, it was observed how the PTG was able to maintain the ability to perform a similar number of repetitions from Set 1 to Set 4, while the CG had a reduction in the repetitions performed. The search for effective strategies for recovery between sets of strength exercises is a constant in recent research [30,31], suggesting that set configuration is a key factor in the regulation of neuromuscular and cardiovascular responses to RT [32].

The study produced some novel findings, specifically, (a) the advantages of PT application between RT sets with regard to the total number of repetitions, and (b) the positive effect on fatigue delay, because there were no changes in EI despite the increase in repetitions. The strict control of the actual repetition velocities performed by the two experimental groups enabled us to isolate the effect of the variable of interest, in this case, velocity loss after the PT application. Current evidence has shown that velocity losses of 30% allow the performance of a high number of repetitions in RT [22,33], which is a key component of muscular performance. In this study, the PTG showed significant differences compared to the CG in the number of repetitions performed (*p* = 0.047), suggesting that PT is an effective technique for improving recovery between sets. Similar methods based on the application of foam rollers have not shown benefits in athletes’ performance when used during pauses between sets [34].

While the neuromuscular processes that cause strength losses in high-intensity muscular efforts are well known, the contribution of each of them in sequences of repeated sets is less clear [35].

The movement velocity during training seems to be a decisive factor in achieving muscle adaptations [36], which occur when performing repetitions at maximum velocity in concentric actions. Adaptation to these actions requires a large amount of power delivered over a short period of time to generate force [37].

The mechanisms by which PT works are not fully understood; changes in fascial components, adhesions, piezoelectricity, myofascial trigger points, and viscoelastic properties of tissue influenced by collagen remodeling and changes in elastin are possible reasons for the ROM and muscle force increases seen after PT [15]. Previous research [15,16] with this technique has focused on the effects on reducing stiffness and increasing range of motion, with a longer application time (2 to 5 min) and lower intensities (1750 rpm).

Loss of velocity is one of the most studied load control indicators in the current research, given the possibility of controlling the level of effort that the athlete will train [7]. Therefore, it is very important to apply the VBT method effectively. It is recognized that when VBT is properly applied, individualization and greater homogeneity of fatigue responses may appear [1].

Therefore, the use of PT may facilitate the maintenance of repetition volumes at a target intensity, which may contribute to hypertrophy [38]. In addition, it can be an option to optimize the time of hypertrophy sessions by being able to reduce the recovery time between sets in order to maintain the total volume of work, given that recovery time is a key factor [39].

One of the most studied aspects in the literature has been the effect of different recovery times on athletes’ performance. Previous research has shown that during RT exercises, a shorter rest interval between concentric and eccentric exercises may result in a higher degree of fatigue, although the total number of repetitions is similar compared to the longer rest interval [40]. This new technique can be applied to minimize recovery times, thus optimizing athletes’ training sessions. As PT is becoming more and more popular among strength and conditioning coaches and athletes, there is a need to determine protocols for use and to know the concrete effects of this novel technology.

As for the limitations of this study, since an intra-group design was not used, it is difficult to draw solid conclusions about the effects of percussive massage treatment. Moreover, no control measure was used for the experimental group, however, all participants were men of comparable strength level, as shown by the results of MV and the total number of repetitions in the first set. However, to our knowledge, this is the first study to apply the PT technique to reduce intra-session bench press fatigue. More studies are needed, with a more robust methodology and applying PT to other muscle groups to corroborate the effectiveness of the PT inter-set rests in improving the performance of resistance training.

## 5. Conclusions

These results provide coaches and fitness professionals with practical information, enabling them to use PT as an effective way to improve athletes’ performance. Currently, there are many strategies for increasing athletes’ recovery during training sessions, and PT may be an effective method. Further research should provide more information about the role of this new technology in the field of RT and help to establish its role in other strength training regimes.

## Figures and Tables

**Figure 1 ijerph-18-07726-f001:**
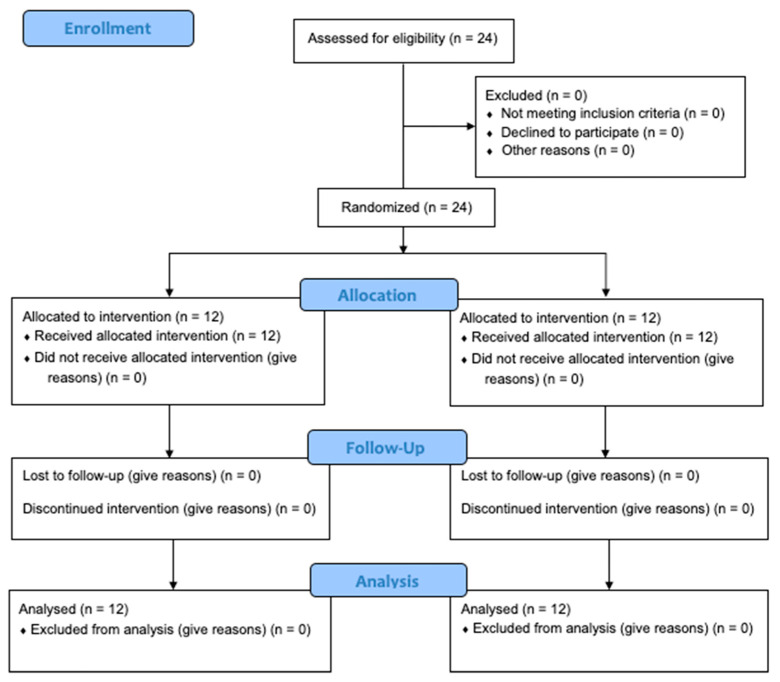
CONSORT diagram.

**Table 1 ijerph-18-07726-t001:** Participants’ characteristics.

	PTG	CG	Total	SD
Age (years)	24.7	23.8	24.3	1.3
Height (cm)	175.3	179.5	177.4	5.6
Body mass (Kg)	78.1	77	77.5	8.3
BMI (kg∙m^2^)	25.5	24	24.7	2.6
RM (kg)	75	78.9	76.9	17
70% RM (kg)	52.5	55.3	54	12

**Table 2 ijerph-18-07726-t002:** Number of repetitions in four consecutive sets performed, applying a 30% velocity loss in the bench press exercise.

Sets	PTG	CG	ES
**Number of Repetitions**			
Set 1	11.4 ± 1.2	11.3 ± 0.9	0.087
Set 2	11.8 ± 1.1	10.7 ± 1.8	0.939
Set 3	11 ± 1.1	9.1 ± 2.3	1.055 *
Set 4	10.3 ± 2.2	8.3 ± 3.4	0.697
Total number of repetitions	44.6 ± 4.8	39.5 ± 6.8	0.867 *

CG: control group; ES: effect size; PTG: percussion therapy group. Note: data are expressed as mean ± standard deviation. Differences between the PTG and CG were analyzed using a Student’s *t*-test. * Indicates significant differences (*p* < 0.05).

**Table 3 ijerph-18-07726-t003:** Movement velocity of the fastest repetition in bench press set.

Variables	PTG	CG	ES *
**Mean Velocity (m·s^−1^)**			
Set 1	0.671 ± 0.094	0.654 ± 0.109	0.167
Set 2	0.631 ± 0.104	0.627 ± 0.100	0.039
Set 3	0.597 ± 0.103	0.591 ± 0.106	0.057
Set 4	0.572 ± 0.102	0.567 ± 0.119	0.045
Set Mean	0.618 ± 0.092	0.610 ± 0.105	0.081
**Peak Velocity (m·s^−1^)**			
Set 1	0.881 ± 0.214	0.882 ± 0.109	0.006
Set 2	0.784 ± 0.188	0.808 ± 0.146	0.143
Set 3	0.715 ± 0.188	0.727 ± 0.127	0.075
Set 4	0.692 ± 0.176	0.663 ± 0.132	0.186
Set Mean	0.768 ± 0.180	0.770 ± 0.134	0.013
**Peak Power (Watt)**			
Set 1	472.2 ± 128.9	457.8 ± 133.9	0.110
Set 2	416.5 ± 105.8	418 ± 118.7	0.013
Set 3	379.2 ± 96.7	374.1 ± 96.7	0.053
Set 4	336.4 ± 86.4	340.7 ± 92.3	0.048
Total	408.6 ± 98.5	397.6 ± 107.7	0.107

CG: control group; ES: effect size; m·s^−1^: meters/seconds; PTG: percussion therapy group. Note: data are expressed as mean ± standard deviation. Differences between the PTG and CG were analyzed using a Student’s *t*-test. * No significant differences (*p* > 0.05) were observed for any variable.

## Data Availability

Not applicable.

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
