# Peer review of "Acute Effects of a Percussive Massage Treatment on Movement Velocity during Resistance Training"

_ijerph, 2021, doi:10.3390/ijerph18157726_

Round 1
Reviewer 1 Report
This is a very interesting article about a massage treatment to improve performance during resistance training.
I want to congratulate with the Authors for the great work: it is written clearly and in a well organized structure.
Follow CONSORT checklist (http://www.consort-statement.org/) is always a good idea when when writing a RCT.
I have only some minor suggestions to help you:
- pay attention to abbreviation (i.e. introduction, line 28: MV?)
- pay attention to English language and style
- in Table 2 I suggest to delete the column with p-value numbers and replace it with asterisks to "shorten" the Table: in this way, it would be easier to read
- please add a phrase about limitations of your study at the end of discussion section
- I suggest the following reference (https://www.ncbi.nlm.nih.gov/pmc/articles/PMC7739672/) about the importance of recovery for performance enhancement, and another one (https://www.mdpi.com/2411-5142/5/4/89) about the importance of new technologies in sports training!
Author Response
Thank you very much for your comments and appreciation.
Best regards

Reviewer 2 Report
Thank you for the opportunity to review this paper and congratulations to the authors of the research idea with high practical implications. Nevertheless, work has great limitations. The main limitation of the study is that no control measure was used for the experimental group. How can the authors be sure that the result is actually a PT effect? maybe the participants in the experimental group were just able to lift the barbell more times than those in the control group? In my opinion, in this case, there should be measurements taken within participants. In my opinion, this disqualifies the article for publication in IJERPH.
Some other concerns:
Introduction:
there is a lack of information on the use of different velocity loss threshold in resistance training. Authors should briefly describe and provide a rationale as well as advantages of this method instead of use velocity to estimate 1RM.
Line 12: „volume of repetitions”? shouldn’t be “number of repetitions”?
Line 28: MV appear the first time, have to be explained. Moreover, this sentence is a bit confusing. In my opinion, it needs to be rewritten
Line 29-36: I am aware that the authors want to present the VBT method, but this information is not related to the purpose of the article. I would suggest that the authors be precise and briefly introduce the reader to the research problem.
Line 38: before the authors used MV why now you introduce average velocity? the reader may think it's something else. Please be consistent
Line 46 and 50: use “velocity” instead of “speed”
Line 54: “inter-set rests” instead of “breaks”?
Line 57: this sentence requires more citations. Does one article show popularity?
Line 67-69: I think that more details about protocols of these studies are needed, e.g. PT application, what groups of participants were studied etc.
Line 70-72: maybe the conditions for regeneration are better, but don't they decrease the capacity of muscle contraction capacity, utilisation of the SSC? reduction of stiffness may prove disadvantageous during high-power/velocity actions. I suggest discussing it, especially since there was no improvement in the study by Konrad et al., 2020 and no effect on muscle strength in García-Sillero et al., seems that might be on velocity contraction but not on force output.
Line 135: what is the rationale for 60%1RM?
Line 144: remove “is”
Line 153: why PT was applied only on the pectoralis? Not in all prime movers during bench press?
Results
I can’t find results for the effort index.
Table 1.
Use dots, not commas. Why there is 70%1RM?
Figure 1.
I think that it would be clearer to show the number of repetitions in the subsequent sets in the table, not in the figure
Discussion
authors should point out that only in the 3rd set there was a significant difference, and try to discuss why there were no differences in the 2nd and 4th set?
In relation to my earlier comments, the authors cannot be sure that the result is an effect of the use of PT, therefore they cannot so emphatically conclude that it is PT that is responsible for the increase in the number of performed repetitions. Moreover, authors should be more specific. PT contributed to an increase in the number of repetitions performed, and not the overall physical performance (if at all), especially that there are contradictory study.
Author Response

(The authors gave the same response as above.)

Reviewer 3 Report
Specific Comments for the Authors
- Abstract and Introduction: while I did not list any specific examples in my comments, please consider a closer look at grammar throughout the entire manuscript. There were not as many concerns in the Abstract and Intro, so I did not list my specific comments as they were hardly noticeable. But the examples that I provide below for subsequent sections should help.
- Page 3, 2.2 Trial Design paragraph 1, line 104: change “heigh” to “height”
- Page 3, 2.2 Trial Design paragraph 1, line 117: reword “to played the best role”, as this is confusing to the reader
- Page 4, 2.3.1 Bench Press Exercise paragraph, line 144: delete “is” before “was calculated”
- Page 4, 2.3.2 Percussion Therapy paragraph, line 147 and line 150: change “he PT treatment” to “The PT treatment” and on line 150 change “at the end of each of the set” to “at the end of each set”
- General Methods question: why was percussion therapy directed at the pectoralis muscles and not the triceps? It seems that you might have gotten better results had you treated the triceps muscles in between sets.
- Table 1: why is 70% RM data listed, and not 60% RM data? I thought that the methods explained (earlier in the text, page 4-line 135) that 60% of 1-RM was used?
- Page 6, Discussion section, first paragraph, line 199: change “study not showed” to “study did not show”
- Page 7, Discussion section, lines 234-235: consider rewording this sentence as it is confusing to the reader
- Page 7, Discussion section, line 241: reword, especially as it relates to your use of the word “gonna”
- Page 7, Discussion section, line 255: change “strength and conditional” to “strength and conditioning”, and reword, especially as it relates to your use of the words “huge gap”
- Overall as it relates to Concluding statements: the study findings are interesting, and may provide a nice initial launching point for other studies to test the value of PT on recovery between RT sets, but you may want to try to be more tempered with our claims of grand findings, as the reported difference of 44 vs 39 reps is a quite small effect size. Your data are worthy because you can see the linear effect of the treatment best with the decline in reps in the white bars and the increase of the reps in the black bars. The lack of effect size and the lack of large numbers of significant differences in your study may be a little disheartening to you, but again, the effect of the treatment appears to be evident with the numbers of reps moving down for no treatment and staying the same (or even going up some) in the treatment group. I can’t help but wonder what you would find with the same study and treatment on the triceps. Nice job overall and good luck
Author Response

(The authors gave the same response as above.)

Round 2
Reviewer 2 Report
Line 103: so this MV refers to movement velocity? which one? Mean or peak? as well as in line 132. I guess to the mean velocity, but it's confusing. In my opinion, you should remove the abbreviation for „movement velocity” and used it in regards to the „mean velocity”. Moreover, which velocity loss? Anyway, this issue needs clarification.
Line 144: pectoralis major? be precise
Line 145: so grip width wasn’t standardized?
Author Response
Dear reviewer,
Thank you for your comments. In accordance with your comments, we have proceeded to make the requested changes.
Thank you very much for your help.
Best regards
